# Bibliometric Analysis; Characteristics and Trends of Refuse Derived Fuel Research

Khadija Sarquah [1,2,*], Satyanarayana Narra [1,3], Gesa Beck [2], Edward A. Awafo [4] and Edward Antwi [1]

[1] Department of Waste and Resource Management, University of Rostock, 18051 Rostock, Germany; satyanarayana.narra@uni-rostock.de (S.N.); edward.antwi2@uni-rostock.de (E.A.)
[2] Berlin School of Technology, SRH Berlin University of Applied Sciences, 10587 Berlin, Germany; Gesa.Beck@srh.de
[3] German Biomass Research Centre, 04347 Leipzig, Germany
[4] Department of Agricultural and Bioresources Engineering, University of Energy and Natural Resources, Sunyani P.O. Box 214, Ghana; edward.awafo@uenr.edu.gh
* Correspondence: khadija.sarquah@uni-rostock.de

**Abstract:** Among the sustainable ways of municipal solid waste management (MSW) is energy recovery, particularly into refuse derived fuel (RDF). However, the potential, application, and research of RDF in existing cases is not exhausted. Additional analysis of literature is needed to provide further insights into the area. The evolution of RDF research over the past 30 years is analyzed and presented in this paper. Using a statistical approach, a bibliometric analysis was conducted for research on RDF from the SCOPUS database to assess perspectives and trends and gain a better understanding of the scope of RDF research. The bibliometric analysis tools, co-citation, keyword occurrence, co-authorship, and bibliometric coupling network, were utilized in VOSviewer to map out keywords, collaborations, and citations. The result from the analysis indicated that scholarly works around RDF were largely written in English (92.7%). Journal articles were the most frequently occurring document type, representing 68.5% of the records, followed by conference papers (24.9%). Out of a total of 1184 documents analyzed, the number of publications rose steadily from 26 in 2003 to 102 in 2021. Most publications on RDF were in the subject area of Environmental Science (648), Energy (483), and Engineering (441). Waste Management, Fuel, Waste Research and Management, and the Journal of Cleaner Productions were the sources that contained most of the publications on RDF research. The contributions (number of publications) in the RDF research were largely from the US (176), Italy (97), Japan (90), Germany (86), China (79), and the UK (74), among others. Collaborations were notable between the US, Europe, and the Asian regions (China, India, and Thailand). Conspicuously missing were research contributions from the African region, comparatively, thus emphasizing the need for contributions from such perspectives. The keyword analysis result further showed studies were within knowledge areas of conversion processes, applications, and management. Thermochemical conversion pathways were highly applied to RDF and thus combustion/co-combustion/incineration (717), gasification (224), and pyrolysis (115). Additionally, research on RDF applications was mostly in the cement industry (150) and electricity generation (55). The time incident analysis outlined recent interest and emerging trends in optimization of valorization processes, towards a circular economy and sustainability. Cross-cutting areas of environmental focus (emissions) were observed along the years analyzed. There is a rising focus on indicators for commercialization, environmental impacts, and optimum production from the analysis. This is useful especially for the emerging regions/territory of research contributions. These discussions would potentially maximize the co-benefits of energy generation and eco-environment sustainability via cost benefits deployments suggested for future research. Analyzing the RDF research trends, these findings are useful for the future endeavor of researchers and developers.

**Keywords:** refuse derived fuel; bibliometric analysis; energy recovery; solid waste management

## 1. Introduction

Among the significant challenges facing urbanization around the world is waste management. Globally, municipal solid waste generation (MSW) is rising. A total of 2.01 billion tons as of 2016 is expected to increase to about 3.40 billion tons by 2050 across various cities worldwide [1]. The effective management of municipal solid waste involves the application of approaches, technologies, and concepts that ensure the protection of public health and the environment. Recovering energy from waste is playing an increasing role in generating "low carbon" energy and meeting targets of energy from renewable sources. In 2018, countries within the European Union (EU) thermally treated 96 million tons of municipal solid waste with energy recovery according to the Confederation of European Waste-to-Energy Plant report [2]. Energy recovery from waste over the years has contributed to the global energy mix. This has contributed to meeting the rise in energy demand as the global population increases with industrialization and economic growth.

One viable and robust form of waste-to-energy from municipal solid waste is refuse derived fuel (RDF). Generally, RDF refers to the segregated high calorific fractions of waste from MSW: household, commercial, or industrial process wastes [3]. RDF from municipal solid waste particularly is sorted portions, which consist of combustible components such as waste plastics, paper, cardboard, textiles, and wood. It serves as an alternative way to reduce landfill waste and energy recovery from municipal solid waste. The utilization of refuse-derived fuel as a renewable energy resource also aligns with the Sustainable Development Goal 7: affordable and clean energy [4].

RDF is widely produced and utilized in most advanced economies such as Germany, Italy, Japan, China, Ireland, the USA, etc. [3]. Interest in energy recovery from MSW as refuse-derived fuel has also been extended to some developing countries such as India, Indonesia, Thailand [5], Mozambique, and Namibia [6]. For instance, about 7.5 MW of electricity is generated from RDF (heating value > 4000 kcal/kg) in India as of 2017 [7]. In the UK, the production of RDF in waste-to-energy facilities has contributed up to a 50% reduction in municipal solid waste being landfilled in the last decade [8]. England also exported 2.6 million tons of RDF in 2019 [9]. Thus, RDF is recognized as a major alternative renewable energy resource.

However geographic distribution and impacts have not been the same. Although technologies exist, waste-to-energy research becomes specific since waste composition, energy needs, economies of scale, and infrastructure differ from one place to the other. Hence, research has not been widely distributed at the same pace. Thus, knowing the status of and various distribution of the scientific information concerning the RDF landscape is of special interest. Both quantitative and qualitative analysis represents an effective methodology to identify which field the research has been mainly directed in and analyze the important links between topics. Currently, bibliometric data analysis is widely used, and it allows objective quantitative analysis. Although this methodology is not new, as its use started back in the 1970s from the Scopus database [10], its proliferation is fairly recent, as seen in many fields on average in the last decade [11]. In bibliometric analysis, the scientific data collected for analysis are codified and ordered and come from global scientific publication databases, e.g., Scopus, Web of Science, PubMed, Dimensions, etc. Bibliometrics is an important tool for determining research trends. Studies with bibliometrics have been conducted in various fields; for instance, in engineering on soil erosion modeling [12] and industrial waste water treatment [13], on circular economy in the building and construction sector [14], and an analysis of the potential uses of brewer's spent grains in a biorefinery for the circular economy transition [15]. In energy and environment, bibliometric analysis has presented research trends in biomass and biofuels [16,17], energy security [18], nuclear energy research [19], and energy research trends from specific geographic locations [20,21], and in social science [11].

The body of literature has grown substantially with the advancement in RDF and its utilization technologies. Various researches have outlined the advantages of RDF over mass MSW, such as higher calorific value; greater homogeneity of physical and chemical

composition; ease of storage, handling, and transportation; lower ash and emissions of pollutants; and reduction of excess air requirement during combustion [22,23]. The amount of waste needed to produce RDF greatly varies, of which the composition of the waste is of much interest. Hence, the quality of RDF can vary widely depending on the waste materials and the extent of processing. Nevertheless, RDF is utilized in energy-intensive fields such as cement, paper, steel, boiler, and metal industry, as well as electricity production and heating. In the EU, about 12 million tons of RDF is reported to be utilized in cement kilns and dedicated RDF incinerators in 2015 [24].

A range of research on RDF has been conducted for contribution in past times. Feasibility studies are required for assessing feedstock suitability as RDF [25,26]. For instance, Gallardo et al. [27] assessed the MSW reject fractions from a biological–mechanical treatment plant in Spain for RDF. The physical and chemical characteristics obtained (classified as Net Calorific Value 2, Cl 2, Hg 5) met quality standards compliance (EN15359:2011) except for Hg. The utilization/end-user processes including thermochemical valorisation processes convert the fuel into useful energy. Aluri et al. [28] performed pyrolysis and gasification experiments on RDF samples. Co-combustion experiments [29,30] conducted on RDF samples resulted in a net gas cost saving of 65 USD/h for cement production. Decision support tools have become inevitable in assessing waste-to-energy systems. Studies such as LCA on RDF production [31] and RDF utilization in clinker production [32] have been explored. Similarly, investigations into techno-economic analysis [33,34] and simulation and preliminary study models [35] have been conducted. These purposely aim at understanding the systems, processes, chemical and physical parameters, costs, and environmental concerns. Chatziaras et al. [36] reviewed alternative fuels utilized in the cement industry including RDF. It was established that fuel substitution resulted in maximum co-benefits of reduced cost and $CO_2$ emissions, while a major concern considered for RDF is to meet limits of chlorine (Cl) content. Gerassimidou et al. [37] reviewed thermogravimetric characterization of RDF and concluded there is high potential to develop TGA-based composition identification. Yang et al. [38] conducted a review on RDF gasification and recorded the production cost of energy estimated at 0.05 USD/kWh. The study also established from the review some countries that replaced coal with refuse-derived fuel reduced $CO_2$ emission by 40% and decreased landfilled MSW by 50%. The study recommended two feedstock co-gasification, as it presents more benefits of minimum tar formation and improved process efficiency. The previous reviews thus have provided insights into specific fields; however, additional analysis of literature using rigorous bibliometric tools can provide further insights not previously grasped due to the vastness of the RDF topic. The use of bibliometric analysis to describe research interests and trends is widespread. However, bibliometric analyses of literature related to RDF are limited. On the other hand, it may be of greater importance to assess literature from a holistic perspective. This would identify the point of development emphasis and the priorities areas of the different aspects of the RDF field. Furthermore, summarizing research trends and current gaps from the multiple perspectives of energy generation, technologies, systems, processes, geographic locations, and others can provide a broad reference for practical implications among various stakeholders. Therefore, it is necessary to assess the development and growth of research in past and recent times. Thus, different from previous reviews, this study contributes to a comprehensive perspective.

Taking this into consideration, the study thus aims to present a quantitative analysis and an overview of the literature on RDF from the Scopus database, specifically considering the following: (1) What is the structure/volume of knowledge and the major key concept that RDF research has explored? (2) Which are the subject areas, journals, institutions, and country/region impacts in the past 30 years? (3) What are the emerging trends for potential future focus? This analysis aimed to describe the research evolution, helpful to predict future scenarios and participation among stakeholders in the sector. As waste has become a valuable resource for energy recovery, particularly for RDF, the results will not only provide an overview of the spot in the specific research area of RDF but will provide useful

information on the broader research area. Thus, assisting researchers in understanding the present condition, predicting the dynamic changes, and serving as a robust roadmap for further investigation in the field of RDF.

## 2. Methods

### 2.1. Data Selection, Matrix, and Scope

The study was guided by the process of bibliometric studies outlined by Donthu et al. [39]. The bibliometric analysis processes consisted of the document selection process, indexed by the SCOPUS database. The advanced search was used to define the field of interest, while words appearing in the title, keywords, and abstract were considered using textual modification. The analysis was based on the search term "RDF", "refuse derived fuel, refuse-derived fuel, waste recovered fuel and Refuse derived fuel*", to include also the derived form, in the title, keywords, and abstract. Some filters were applied for a more pertinent selection of the articles, such as the period range. Non-English publications were excluded during the advanced search. The bibliographic search was conducted in November 2021, illustrated in Figure 1. The data were further reduced from the exclusion of non-related keywords in the search string, elimination of duplicate entries, or referring to the same document or incomplete/incorrect bibliographic details. This include works directly on the anaerobic digestion process, biogas, and compost from MSW since a few of them referred to keywords such as "waste-derived fuel". The field was limited to 30 years (1991–2021). The data was exported in the .csv file format for analysis in Microsft Excel and Vosviewer.

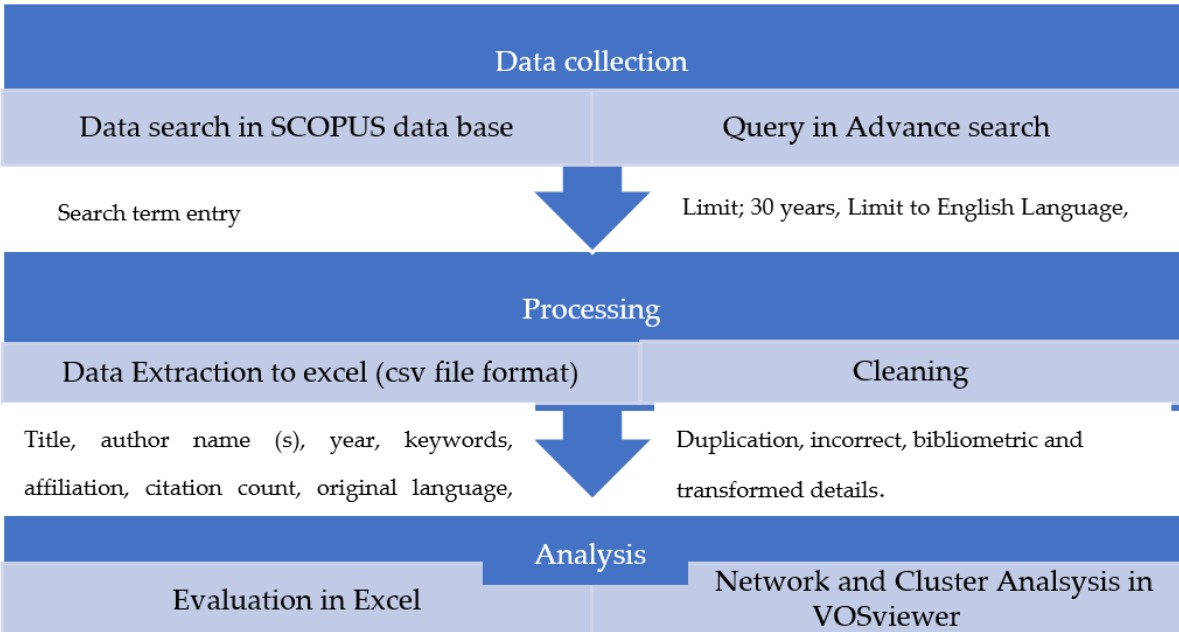

**Figure 1.** Flow of method and analysis.

### 2.2. Analysis

Aspects considered for analysis include the quantitative: areas of research, years and relating number of publications, countries/origin of studies, language, journal distribution, authors, and institutions. The qualitative analysis considered the thematic areas and keyword/term mapping. The bibliometric analysis explored both Excel and VOSviewer for both the quantitative and qualitative. VOSviewer is an open-source software for network and visualization analysis. Data cleaning in excel was performed to remove duplicates, messy data, and transforming formats. The graphs/tables were generated in Microsoft Excel using the output data from VOSviewer. Word frequency analysis is a key method of content analysis [39]. This was performed in VOSviewer using the co-occurrence analysis

tool. Keywords and expressions indicate the core content of the literature as the research object. In this way, the trends and changes in scientific research of a certain area can be analyzed quantitatively. According to Donthu et al., the author keyword analysis helps to know clearly about the research hotspots in the field. Author keyword co-occurrence analysis shows the interaction between the various research directions. Combining author keyword frequency and author keyword co-occurrence analysis is a useful and effective method for content analysis, which not only can have a holistic understanding of the research field but also suggests the future research direction. Co-authorship, co-citation, and bibliometric coupling were also utilized in VOSviewer.

The h-index, or Hirsch index, is an impact matrix, proposed by Hirsch to measure the productivity and impact of published works of particular scientists, scholars, institutions, or countries. According to Hirsch [40], a scientist has index H if H of his/her Np papers have at least H citations each, and the other (Np-H) papers have no more than H citations each, where Np is the number of papers published over n years. Thus, H-index, measures for quantity (i.e., number of publications), and impact (i.e., number of citations) are integrated into a single indicator. This study considered the h-indexes of research scholars. Statistics on authors usually concentrate on the change of the number of outstanding and ordinary authors over time. Citescore, SJR, and SNIP were also considered for the source analysis.

## 3. Results and Discussion

### 3.1. The Characteristics of Research

A total of 1184 documents related to refuse derived fuel research over the past 30 years from the Scopus database were considered. It was observed that research on RDF could be traced back to the 1970s. The research was largely reported in the English language (92.7%). Other languages include Chinese, Japanese, German, Polish, Italian, Korean, and French. Applying the advanced search resulted in 1184 publications considered for the analysis. The obtained data were categorised into different document types. Journal articles and conference papers were predominant and accounted for 68.5% and 24.9%, respectively. Review papers accounted for 3.1%, book chapters 1.9%, conference reviews 0.5%, books 0.2%, and others 0.8%. The volume of work did not have a constant pattern as shown in Figure 2. Despite the fluctuations, there was a steady growth from 43 in 2010 to 102 in 2021. One possible reason for the rise in research was the continuous advocacy for sustainable waste management and landfill issues. As such, waste-to-energy is recognized as a sustainable waste management option and well as GHG emission reduction from landfills. The number of citations also fluctuated; nevertheless, there was a significant rise with 1385 and 406 citations being the highest recorded in 2015 and most-cited paper, respectively. Comparatively, recent works have a smaller number of citations, which is attributed to the time taken to accumulate citations relative to earlier publications. The gradual increase of publications in the last two decades represents a growth trend in interest in the field and research communication in the RDF area.

### 3.2. The Subject Area Covered

The output of the research from RDF was covered in 21 subject areas including energy, engineering, material science, chemistry, computer science, business, and economic and social sciences. The top six subject areas in terms of the number of annual and total publications are shown in Figure 3. There is a record of growth within the subject areas in recent times, with the highest (54) recorded in environmental sciences in 2021 with a total of 648 records. Relatively, other bibliometric studies on energy resources from waste perspectives recorded subject areas such as environmental science as a focus [38,41]. This is also an indication of energy recovery from waste being rated highly as an environmental mitigation measure.

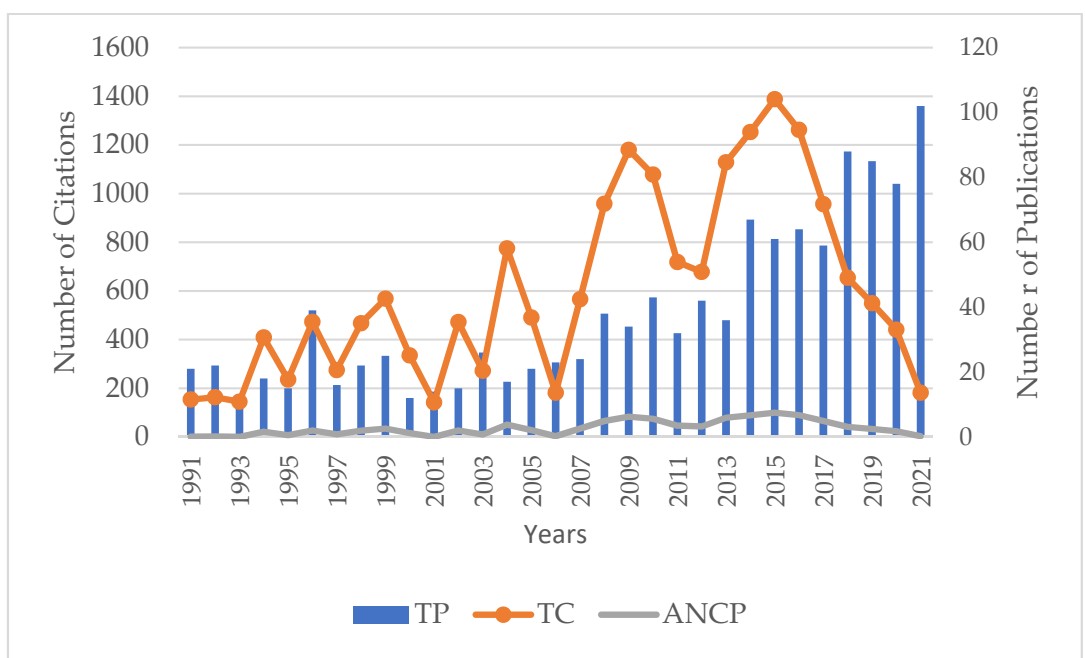

**Figure 2.** Trends of publications (TP: total publications, TC: total citations, ANCP: annual normalized citation per publication).

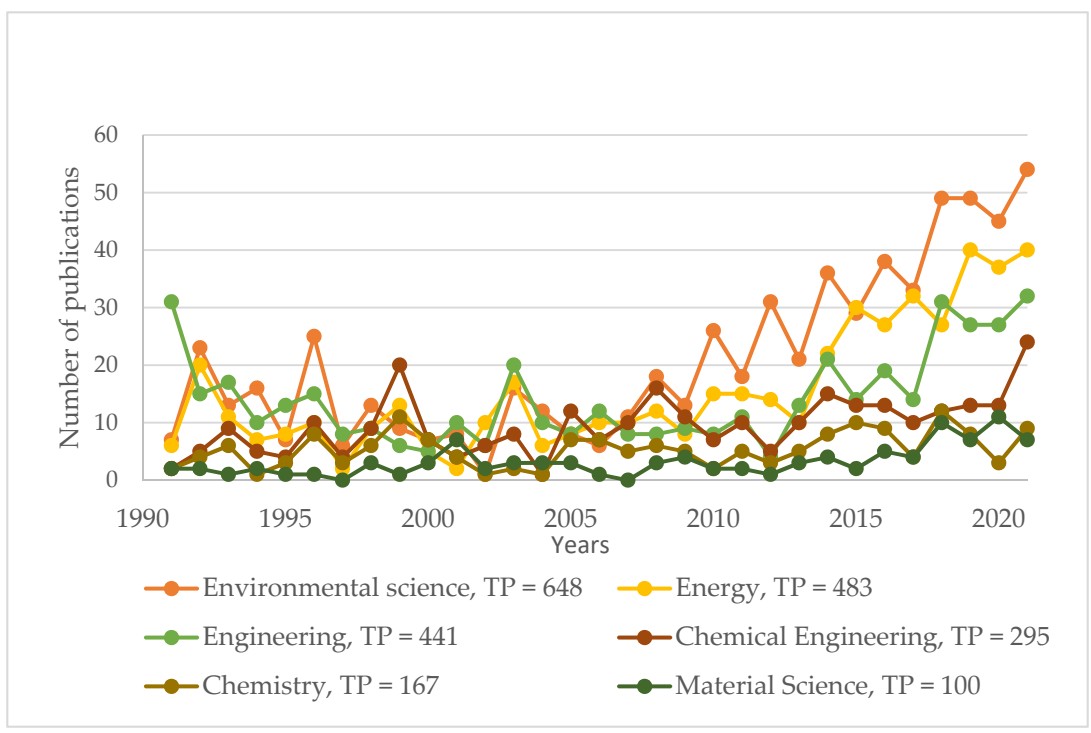

**Figure 3.** Number of publications within the different subject areas (1991–2021).

### 3.3. Research Concept Analysis

### 3.3.1. Keyword Analysis

The key concepts and hotspots that have been explored by researchers were investigated. Keyword analysis was used to explore the knowledge structure and describe the emerging trend and main topics within the research area [39]. VOSviewer was employed for this analysis using the co-occurrence of keywords tool. According to a review presented by Alemán-Nava et al. [21], keywords represent the terms that capture the significance

of the document and search and identify the trends in the various branches of science. Cluster analysis was undertaken to identify main research streams and trends. A total of 2237 author keywords were obtained from publications. Most of the keywords (78.3%) appeared only once, which indicated the wide area of the research priorities/disparity. About 21.6% appeared more than twice, while 10.3% of the keywords appeared a minimum of three times and correlated. This indicated that they were related to the main research streams of RDF.

As seen in Figure 4, the clusters are positioned close to each other, while some overlapped, showing a closer association of keywords forming the clusters. The knowledge structure from the publications was mapped onto four clusters, two major clusters and two surrounding clusters. These include conversion processes, production, material and product characteristics, and RDF application and management. This falls in line with the perspective of RDF (production, application, management, and economics) shared in the report of the European Commission [3].

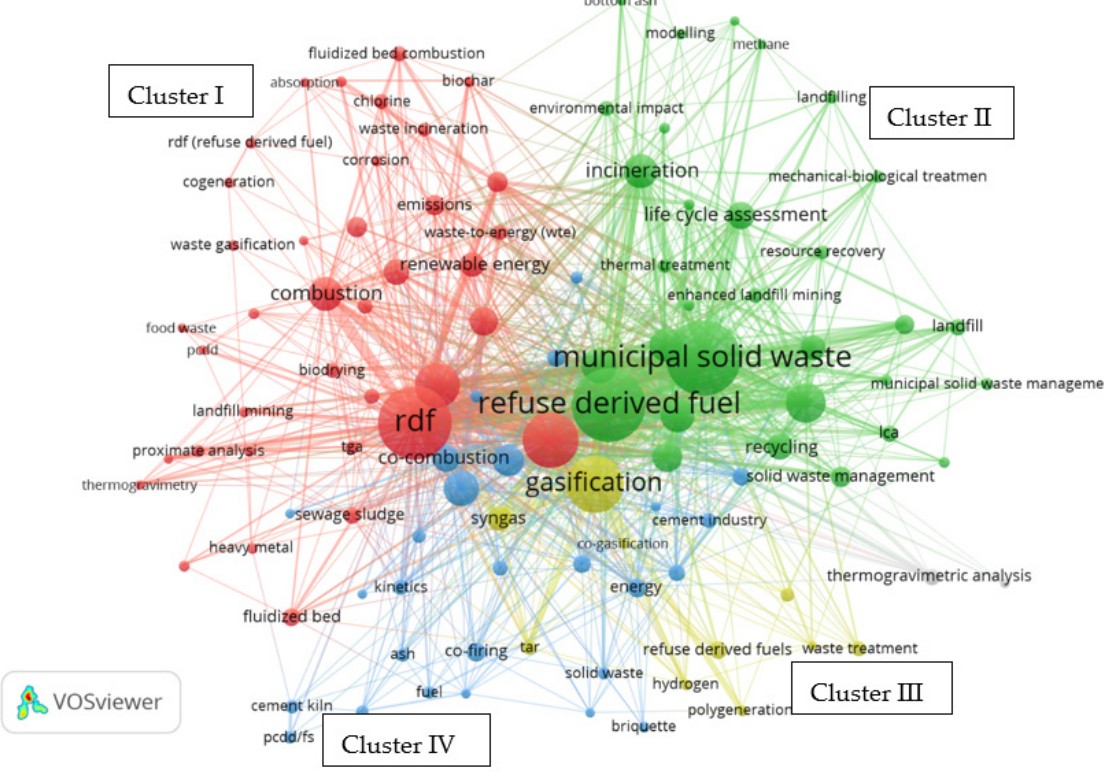

**Figure 4.** Keywords analysis (a minimum of three occurrences of keywords was applied to the co-occurrence network, and 115 out of 230 met the threshold).

Largely, the most occurring keywords were "refuse derived fuel", "RDF", "refused-derived fuel", and "refuse derived fuels" (1360; 59.2%) being the main search term, followed by "municipal solid waste*" and "MSW" (569; 25.4%) in Cluster I and Cluster II. MSW represents the main sources of RDF production. A report on RDF from the European Commission [3] highlighted that sources of RDF include municipal solid waste (MSW) and commercial or industrial process wastes. On the other hand, publications of RDF from sources such as sewage sludge, waste tyres, etc., were recorded. Physical pre-treatment such as pelletization is applied to RDF to improve density and volume for transport as it is usually composed of very lightweight materials, 100–150 kg/m$^3$ [42]. The focus areas are noticed in the cluster with bigger nodes. Mechanical Biological Treatment (MBT), material recovery recorded from research, represents the process of obtaining RDF. The main conversion pathways from the RDF research are thermochemical conversion, combustion/incineration, gasification, and pyrolysis in Clusters I, II, IV. The most frequently

used author keywords reflect a large portion of literature into such process as developing effective methods for RDF utilization. The keywords "combustion", "co-combustion", "incineration", and "co-incineration" are frequently occurring (717), and gasification (224) and pyrolysis (115) are conversion processes from the keywords mapping. In practice, it is also a widely practised form of the RDF valorization pathway; a matured technology as reviewed in the study [43]. Publications from gasification and pyrolysis recorded present an alternative to conventional combustion/incineration processes. This is consistent with findings from other research [33,44–46]. Thus, the production and use of syngas and pyrolysis oil are considered a relative upper hand considering the product gains and environmental cases. Three clusters contained research keywords related to thermal treatment technologies: combustion, gasification, and pyrolysis. Treatment methods such as torrefaction, carbonization, and hydrothermal carbonization received relatively fewer research records for RDF within the period.

In Clusters I, III, and IV, the author keywords "cement industry", "cement plant", "cement kiln", and "cement manufacturing" appeared frequently (150) within the period. This shows affirmation to the RDF utilization mostly in energy-intensive units. In formation, most of the feasibility studies recorded such as [26,47–49] were regarding fuel substitution and utilization in cement kiln. As such, it is preferred to sort MSW into RDF components due to quality and environmental concerns. MSW is used in cement kilns after sorting and balling into RDF [3]. Electricity generation is another aspect of RDF utilization recorded from the publications (81). Another area of RDF application is in fuel cells, however, not a majority of research recorded. Fuel cells particularly the direct carbon system utilize carbon-rich material as fuel to produce electricity from the chemical energy stored in the fuel directly. On the other hand, fuel cells can also utilize syngas or hydrogen from RDF. Although these applications have not received many implementations, it is highly considered the next generation [33]. There are reported studies capturing high energy density, thermal efficiency during phase change, and safety [33,50].

In addition, the production of other materials such as hydrogen, adsorbents (activated carbon and RDF char), and charcoal from thermolysis has also received some research and development in the area of RDF. It is considered a key area of material recovery either from direct RDF or by-products such as ash from its utilization. Adsorption is employed for the remediation of wastewaters containing dyes, heavy metals, etc. This has gained increased attention as a treatment because it is a very efficient technique that operates at ambient temperature and pressure, whilst generating low amounts of residues [51,52]. As studied by [43], RDF that does not favour energetic valorization (low apparent density, low calorific value, high ash content, and high chlorine content) produced quality char (95.9% removal of the dye) as adsorbent. Alternative material recovery from RDF as such is also gaining ground in recent times from the study, which also aligns with feasibility results obtained by Porsnovs et al. [53]. As a result, the approaches for dedicated incineration plants, heating systems, cement kilns, electricity generation, and CHP/cogeneration systems have been priority areas of research for many jurisdictions. The management aspect of RDF research comprises environmental, emissions, by-products use/disposal and economic highlights in Clusters I, II, and IV. Records of management concepts, LCA: life cycle assessment (135), economic/cost analysis (96), and emissions, were obtained from the keyword mapping. LCA is considered one of the decision tools for waste-to-energy systems, whereas techno-economic assessment informs investment decisions. These are decision support tools utilized in determining feasibility and sustainability of waste-to-energy systems. Areas of policy and regulations are considered factors that facilitate waste-to-energy system implementation [54], including RDF. These far advanced with established structures in most advanced economies utilizing RDF [55]. On the contrary, it is reported as a major setback in implementing waste-to-energy in some developing, African and marginalised countries, since most policies' objectives are unclear towards a particular roadmap as assessed in Indonesia and brazil [56,57]. This, however, explains the lesser research records identified in that aspect from this study.

3.3.2. Period Analysis

The keywords incidence in years, from 2010 to 2020, is shown in Figure 5. It was observed that there is a transition from production to environmental and management focus in research. Recent areas from the analysis of keywords used include the circular economy, sustainability, life cycle assessment, RDF char, torrefaction, carbonization waste-to-carbon, and zero waste. This possibility is attributed to more management, environmental concerns, climate change, and sustainable resource use [58]. It also shows environmental concerns of processes regarding waste-to-energy has been of great interest as presented in the bibliometric study [38]. Studies regarding emissions, pollutants, corrosion, and dioxins have been considered throughout the study period. The emerging research on improving the physicochemical, mechanical, and energy properties of produced RDF include biodrying, torrefaction, and carbonization as obtained in Figure 5. The use of torrefied RDF in pyrolysis and co-combustion is also reported recent. There is increasing diversity in research methodologies and focus. On the contrary, there are recent studies emerging from Africa and other developing countries [29,30,59–62] with new and holistic methods of approach. Thus, a holistic approach towards sustainable waste management.

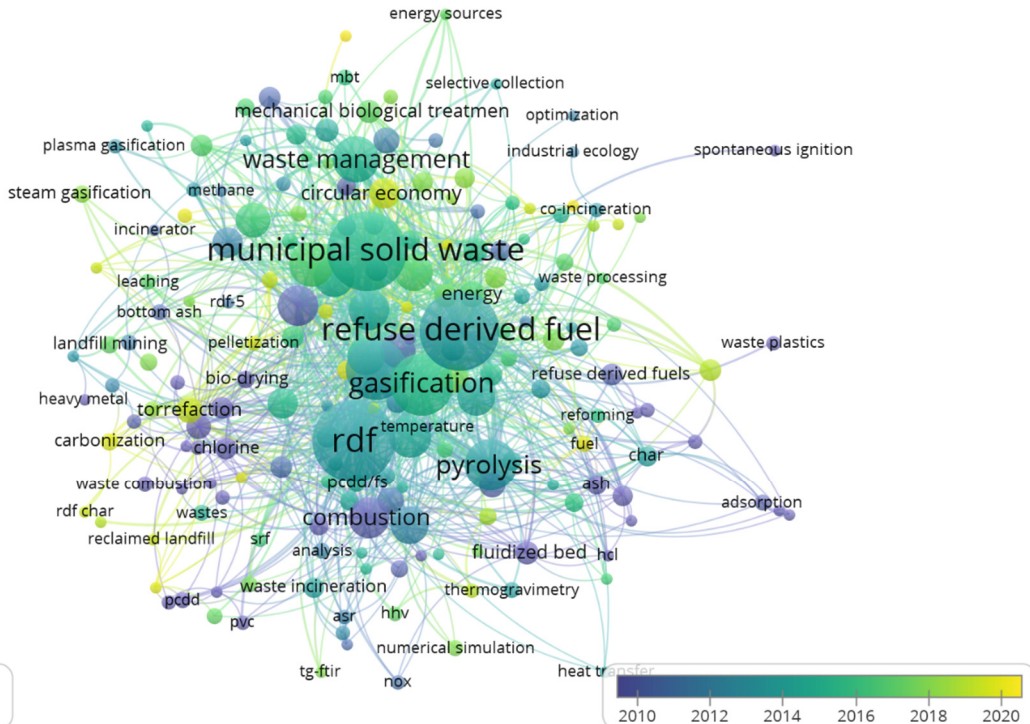

**Figure 5.** Keywords incidence in years.

*3.4. Contribution*

3.4.1. Country/Territory Performance

Contributions from countries/territories to the body of knowledge in the area of RDF as found out from this study is presented in Figure 6. High publication counts were recorded from the United States, Europe (Italy, Germany, UK, and Poland), and Asian countries (India, China, Japan, and Thailand). A total of 1174 publications with author information were obtained from 74 countries that have contributed to this area. There was a limited contribution to the literature from Oceania and African countries; none were among the top 10. Considering developing countries in general, an appreciable number of works were reported from India (51), Thailand (48), Indonesia (41), South Africa (8), Jordan (7), and Morocco (2). This shows that research interest is growing, which is encouraging. However, studies from some African countries (Algeria, South Africa, Morocco, and Nigeria, were recorded in recent times (2015 to 2021), representing a contribution of less than 1%. This

is in line with the GIZ report as waste-to-energy systems are limited in most developing countries with very few successful cases [63]. Thus, the need for more studies on RDF from non-western countries' perspectives is more pronounced.

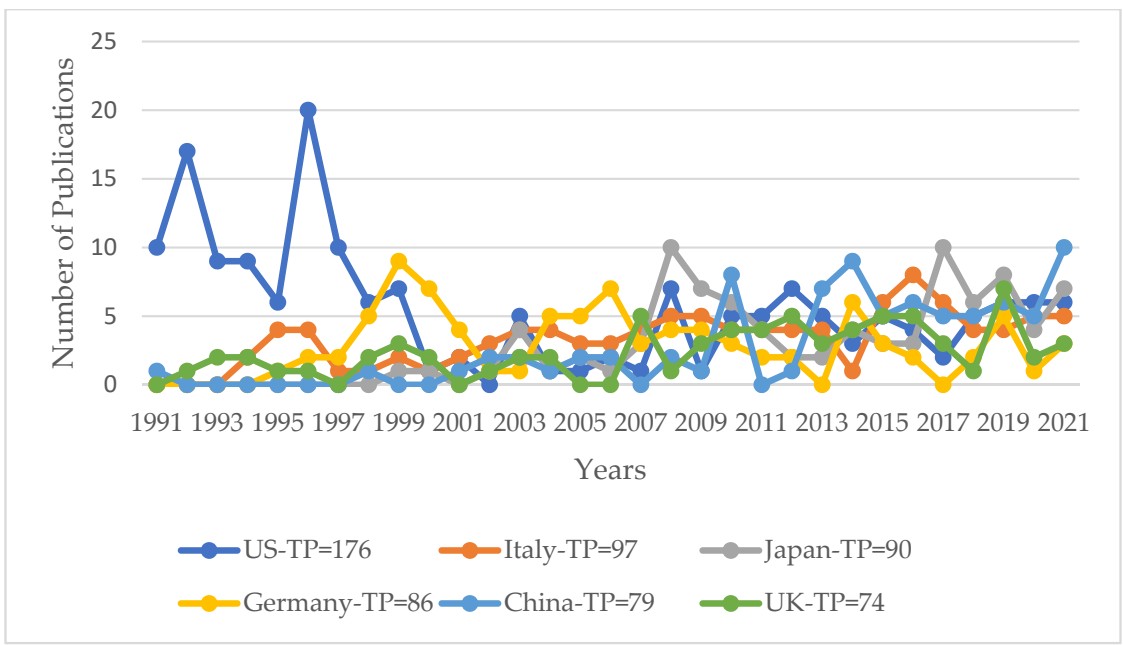

**Figure 6.** Number of publication by country.

The time-trend analysis of the 6 most productive territories is presented in Figure 6. Observably, the US had a leading position during the period of analysis, although publications declined along subsequent years. Similarly, the research growth trend rate was high in other developed countries such as Italy, Germany, the UK, and China. Arguably, one possibility of contribution is from the waste-to-energy policies and strategies instituted by advanced economies such as in the EU. The introduction of regulations in the European Union directives such as zero landfill within the EU strengthened the research in the area of RDF as an alternative fuel and its valorization. As such, energy recovery and GHG emissions reduction from waste draw more attention in these countries. In 2005, Germany adopted a ban on the landfilling of recyclable and organic waste, leading to overproduction of RDF as well as the cement sustainability initiative [64]. On the contrary, research interest observed from China, India, Indonesia, and other Asian countries could be driven by the high population density and the need to find solutions to waste management and the experiences of difficulties to locate suitable sites for landfills [38]. This has motivated more research on MSW and residual waste as alternative fuels.

3.4.2. Collaboration Analysis

Country collaboration analysis was performed in VOSviewer using the co-authorship tool. A total of 64 countries had engaged in collaborative works from the country co-authorship mapping, as shown in Figure 7, forming seven clusters with 179 links. In Cluster I, the UK was central among 14 nodes and 17 links between countries or territories around the world in RDF-related research. Sweden, Italy, and China had the closest collaboration, while others included Canada, Spain, Botswana, and Morocco. Clusters II and IV were largely made up of Asian and European countries. China and Italy were the central nodes in Cluster II. A total of 11 links of collaboration were observed between Italy and countries such as Romania, Spain, Germany, Austria, Portugal, Belgium, Netherlands, Taiwan, Sweden, and the UK. Among these, Romania and Italy had the largest collaboration with a link strength of seven. China had a total of 19 links with other countries such as Germany, Japan, and Finland. The central node of Cluster IV (14 links) was Japan, a strong

collaboration with Thailand and Indonesia. The US was central in Cluster III, with a total of 10 nodes and 23 links of collaboration. Strong links also existed between Poland (nine), Japan (five), and Greece (five). Others were between US and Germany, Thailand, and China. Germany had collaborations existing between Austria, China, Brazil, the Czech Republic, Jordan, and Algeria. India was central in Cluster VI and had 10 links of collaboration, while Sweden in Cluster V exhibited 14 links of collaboration.

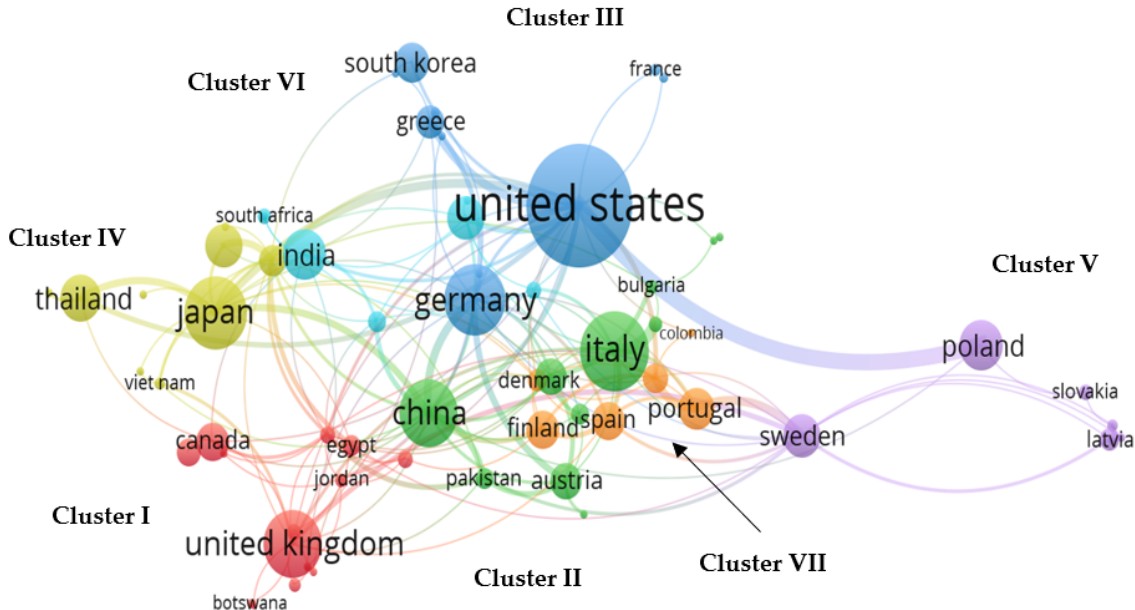

**Figure 7.** Collaborations among countries based on publications (co-authorship analysis; a minimum of 1 was applied to the number of publications by a country. A total of 64 out of 74 met the threshold in VOSviewer displayed).

Generally, the majority of collaborations were among European countries as well as between Europe and the US. This was observed in other studies such as in food waste research [65], biomass analysis [17], and education [11]. Some collaborations also existed between Asian and European territories. This justifies a lot of research and practice within waste-to-energy originating from these areas [13]. EU specifically consumed RDF of about 5 million tons in a cement kiln according to the European cement Association in 2015 [24]. India generates about 7.5 MW of electricity from RDF [7], while about 13 RDF-based facilities exist in the US [5]. Thus, the correlation between the research output and RDF production (implementation) is strong, showing a higher performance of academic influence and active international cooperation. The top 10 countries are coherent in maintaining considerably high levels in both parameters since there is the dedication to research resources and, at the same time, develop and install systems in that regard. This explains their tags as leads in the production and use of RDF [6] and thus the production of technology and scientific developments in the area.

Nevertheless, the possibilities of RDF production and utilization in currently non-utilising countries worldwide exist. In the case of the sub-Saharan African region, there are large potentials for RDF as a result of growing MSW generation and industrialization, whereas the prevalent form of MSW management is landfilling. Thus, RDF could contribute to meeting industrial energy needs and the energy poverty gap with decentralized systems [66]. For instance, in Nigeria, Kenya, Ghana, Togo, Uganda, and Algeria, where cement production is growing [67] with less investment in waste-to-energy [68], there exist large potentials for RDF utilization. Among the challenges facing the local cement production industry include high energy costs [67,69]. However, there are success stories from Namibia in terms of RDF utilization in cement production to learn from [70]. An

investment in research of RDF is significant to bridge the gap, while it will also complement efforts at implementing waste management strategies with a focus on waste-to-energy.

### 3.5. Analysis of Sources, Authors, and Institutional Contributions

The source type of the publications was journals and conference proceedings. The data obtained presented a total of 133 different journals for studies related to RDF from 1991 to 2021. The publications were distributed across a range of journals. The list of the top six journals having above 20 publications is presented in Table 1. Analyzing the journals also showed their origin as from the UK and the Netherlands, representing a dominating status of developed countries in the sources. A maximum of 87 publications and a minimum of two publications were recorded from the sources. The Waste Management journal recorded the highest publications on RDF, cited by 751 other studies, being the most influential journal in RDF research.

**Table 1.** Characteristics of top journals of the publication analysed.

| Journal | TP [a] | TC [a] | CiteScore [1] | SJR [1] | SNIP 2020 [1] |
|---|---|---|---|---|---|
| Waste Management | 87 | 751 | 11.5 | 1.81 | 2.23 |
| Fuel | 45 | 634 | 9.8 | 1.56 | 2.01 |
| Waste Management and Research | 32 | 205 | 4.6 | 0.71 | 1.07 |
| Journal of Cleaner Production | 30 | 466 | 13.1 | 1.94 | 2.48 |
| Energy and Fuels | 24 | 510 | 6.1 | 0.86 | 1.16 |
| Fuel Processing Technology | 24 | 272 | 11.3 | 1.49 | 1.85 |
| Energy | 21 | 340 | 11.5 | 0.86 | 2.01 |

[a]: from this study, TP: total publication, TC: total citation, [1] 2020 figures from Scopus data base, SNIP = source normalised impact per paper; SJR = Scimago journal ranking.

Analysis of affiliations based on author information resulted in a total of 160 institutions, comprising 2291 sub institutions (department/faculty/center) has contributed to the body of RDF research in both as collaborations and as single institutions. Only 159 institutions were registered with two or more publications. The institutions were distributed among the countries/regions of contributions in Table 2, where countries and their top five contributing institutions are reported.

**Table 2.** Country and top five institutional contributions.

| Country | Institutions | TP |
|---|---|---|
| US | Iowa State University | 13 |
| | Western Kentucky University | 9 |
| | Columbia University | 9 |
| | Wrocław University of Environmental and Life Sciences | 9 |
| | University of Florida | 5 |
| Italy | Università degli Studi di Roma Tor Vergata | 10 |
| | Sapienza Università di Roma | 10 |
| | Università di Trento | 9 |
| | Consiglio Nazionale delle Ricerche | 9 |
| | Politecnico di Torino | 8 |

**Table 2.** *Cont.*

| Country | Institutions | TP |
|---|---|---|
| Germany | Technische Universität Dresden | 10 |
| | Technical University of Berlin | 7 |
| | Ruhr-Universitat Bochum | 6 |
| | Vecoplan AG | 5 |
| | Hamburg University of Technology | 5 |
| Japan | Nagoya University | 15 |
| | National Institute for Environmental Studies of Japan | 10 |
| | Kyoto University | 8 |
| | National Research Institute of Fire and Disaster | 8 |
| | Toyota Motor Corporation | 7 |
| China | Zhejiang University | 18 |
| | State Key Laboratory of Clean Energy Utilization | 17 |
| | Shenyang Aerospace University | 9 |
| | Wuhan University of Technology | 7 |
| | Ministry of Education China | 6 |
| UK | University of Leeds | 23 |
| | University College London | 8 |
| | Advanced Plasma Power Limited | 5 |
| | The University of Sheffield | 4 |
| | The University of Manchester | 4 |

A total of 3371 authors have contributed within the period of analysis. A total of 2702 authors had at least a citation recorded. The h-index and affiliation information of authors were obtained from the Scopus website [71]. The first eight authors with at least 10 publications are recorded in Table 3.

**Table 3.** First eight profiling authors with 10 or more publications in RDF research (1991–2021).

| Author Name | TP [a] | TC [a] | h-Index [1] | Current Affiliation [1] |
|---|---|---|---|---|
| Williams Paul T | 21 | 912 | 80 | University of Leeds, Leeds, United Kingdom |
| Chang Nibin | 13 | 237 | 47 | University of Central Florida, Orlando, United States |
| Vilarinho, Cândida | 13 | 47 | 11 | Universidade do Minho, Braga, Portugal |
| Mori, Shigeikatsu | 12 | 152 | 25 | Nagoya University, Nagoya, Japan |
| Białowiec, Andrzej | 10 | 80 | 15 | Wrocław University of Environmental and Life Sciences, Wroclaw, Poland |
| Chang, Ying Hsii | 10 | 207 | 10 | Industrial Technology Research Institute of Taiwan, Hsinchu, Taiwan |
| Gonçalves, Maria Margarida | 10 | 42 | 18 | Faculdade de Ciências e Tecnologia da Universidade Nova de Lisboa, Caparica, Portugal |
| Nobre, Catarina | 10 | 41 | 5 | Collaborative Laboratory for Biorefineries, Sao Mamede de Infesta, Portugal |

[a]: Analysis from this study, TP: total publication, TC: total citation, [1] From Scopus database.

### 3.6. Citation Analysis

The highly cited publications in RDF research (1991–2021) were analyzed. Terms such as total citations (TC), territory/area research was conducted, annual average, source, and year of publication (YP). Generally, the Waste Management journal had a majority of these works, while they were authored from the European territory. Among these, in Table 4, it was observed that the most cited research was published in 2009, with 406 recorded citations. The publication was authored from institutional collaborative work (Italy and Austria) published in the Energy journal by Elsevier with 11.7 average citations per year. The study through a life cycle multi-method multi-approach analysed and compared different waste management scenarios. The study concluded that sorting plants with RDF and material recovery presented the best option based on energy output and global warming potential among other options of landfilling and mass burn. The second most-cited publication reviewed thermal treatment systems with energy recovery among different waste fractions. RDF was reported to be utilized through incineration, gasification, plasma gasification, and pyrolysis in various plants. Among the highly cited publications were the conversion process, environmental impact, and evaluation, which are also among the research hotspots in the RDF field in recent years. RDF from MSW is a valuable alternative fuel and resource for the production of other materials, chemicals, and by-products. The research trends also illustrated re-use strategies for the production of higher value and marketable by-product use as a sustainable raw material. As explained by Donthu et al. [39], the variations in the number of citations of research work can reflect the impact of the publications and determine the intellectual structure of the research field.

**Table 4.** Highly cited research.

| Year | Title | TC | TC/yr | Source | CA |
|------|-------|----|-------|--------|-----|
| 2009 | Life cycle assessment (LCA) of waste management strategies: Landfilling, sorting plant and incineration | 406 | 11.7 | Energy | Austria, Italy |
| 2015 | A review of technologies and performances of thermal treatment systems for energy recovery from waste | 274 | 12 | Waste Management | Italy |
| 2014 | Pyrolysis technologies for municipal solid waste: A review | 238 | 12.8 | Waste Management | China |
| 2016 | Waste-to-energy potential: A case study of Saudi Arabia | 205 | 10.4 | Renewable and Sustainable Energy Reviews | Saudi Arabia, Jordan |
| 2009 | Pyrolysis kinetics and combustion characteristics of waste recovered fuels | 185 | 5.3 | Fuel | Greece |
| 2010 | Fuzzy multicriteria disposal method and site selection for municipal solid waste | 176 | 7 | Waste Management | Turkey |
| 2004 | Adsorption of phenol and reactive dye from aqueous solution on activated carbons derived from solid wastes | 169 | 3.7 | Waste Research | Japan and Thailand |
| 2005 | Alternative strategies for energy recovery from municipal solid waste: Part B: Emission and cost estimates | 167 | 7.1 | Waste Management | Italy |
| 2002 | The influence of biomass temperature on biostabilization-bio drying of municipal solid waste | 157 | 7.8 | Bioresource Technology | Italy |
| 2007 | Characterization of products from the pyrolysis of municipal solid waste | 153 | 6.5 | Process Safety and Environmental Protection | UK |

TC/yr; total citation per year, CA; country of authors.

## 4. Conclusions

Bibliometric and network analysis of literature in refuse-derived fuel research from the Scopus database is presented. The data and analysis showed research within the refuse-derived fuel sphere stems back from the 1970s. Much of the development was found to be contributed from the US, China, Germany, and India. There are comparatively fewer contributions from the African region. Several locations within the sub-Saharan region are underdeveloped in terms of this theme, as issues on waste management and waste-to-energy become contextually driven. The study also showed that the cooperation of countries was limited to specific territories: the US, Europe, and some Asian countries. This shows the need for improved efforts towards research collaboration within the field. This would improve the limited research contribution from especially developing countries and related issues of implementation. International collaborations also unveil opportunities for capacity building and technology transfer. The key journal sources of publications on refuse derived fuel included Waste Management, Fuel, Waste Management and Research, and the Journal of Cleaner Productions. Meanwhile, research was within the subject areas of environmental science, energy, and engineering. The thematic scope of the analyzed publications was diverse: conversion process, utilization, and management. Popularly, research in RDF application was in the cement process and electricity generation. Although a lot of research has been undertaken decades ago, there is still the need for advancement in processes, applications, and prospects. Few studies have been reported on thermochemical upgrading of RDF, material recovery (e.g., hydrogen, as adsorbent, use of ash) other than energy from RDF and LCA as the main research focus. Furthermore, circular economy and sustainability are in recent times highly rated and have become the direction for the most research. In times of diverse growth in technologies and advanced knowledge, AI and IoT may be possible for applicable future research needs or reshaping the existing research.

The review, therefore, foresees prospects are directed at emerging African countries for research and implementation. To this, research on modeling and simulation of elements for multivariable decision making and planning for scenarios while incorporating optimization of processes is highly anticipated. One aspect that is also relevant for a transition towards RDF as an alternative fuel is to explore what infrastructure, policy, and regulations have in terms of use and industry example in a symbiotic approach in most African countries.

The future research and applications of refuse-derived fuel on a global scale appears a promising prospect considering works, experimental and industrial output, and solving environmental challenges of waste management. Towards the analysis of trends, this study would assist to identify the stance in the RDF research and determine projections/tendencies on complementary topics in the field. Refuse-derived fuel is considered as playing an important role in not only improving the energy mix, meeting energy demands, and reducing landfill but also in combating climate change.

**Author Contributions:** Conceptualization, K.S.; methodology, K.S., E.A.; validation, K.S., E.A., G.B., S.N. and E.A.A.; formal analysis, K.S. and E.A.A.; data curation K.S., S.N., G.B., E.A.A. and E.A.; writing—original draft preparation, K.S.; writing—review and editing, K.S., S.N., G.B., E.A. and E.A.A.; visualization, K.S.; supervision, S.N., G.B. and E.A. All authors have read and agreed to the published version of the manuscript.

**Funding:** The study was supported by the BMBF (German Federal Ministry for Education and Research) within the project "Waste to Energy: Hybrid Energy from Waste as Sustainable Solution for Ghana" (03SF0591E).

**Institutional Review Board Statement:** Not applicable.

**Informed Consent Statement:** Not applicable.

**Data Availability Statement:** Not applicable.

**Conflicts of Interest:** The authors declare no conflict of interest.

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
