# Peer review of "Bibliometric Analysis; Characteristics and Trends of Refuse Derived Fuel Research"

_sustainability, doi:10.3390/su14041994_

Round 1

Reviewer 1 Report

The study and manuscript are generally good.  However, some analyses and deductions are erroneous and require modification. Corrections and modifications below:

  1. Line 76 – 7.5MW. Include energy value not just power.
  2. Section 2.1 – is this an established methodology? Give references and explain how it has been used by others, especially in areas related to sustainability.
  3. Line 131 – erroneous cross reference.
  4. Line 146‐147 – Needs rephrasing.
  5. Line 181 ‐ from 43 in 2010 to102 in 2021.
    Missing space.
  6. Line 185. Is the number of citations (406) correct? It does not correspond to Figure 3.
  7. Line 198 – error in cross reference.
  8. Line 252‐253 – Sentence needs rephrasing.
  9. Line 268 – check spelling.
  10. Lines 303‐305 need rephrasing
  11. Lines 311‐315. The description is explaining the cluster plot in Fig 8 not the graph of Fig 7.
    The graph shows that UK was the country publishing the most. The explanation does not correspond.
  12. Figure 7: are values correct?! UK > US?
  13. Line 328‐330. The statements do not correspond to the graph.
  14. Fig 8 – Some clusters are not numbered.
  15. Line 375‐376 – sentence needs rephrasing.
  16. Line 402‐3 needs rephrasing.
  17. Line 423 – spelling

Author Response

Response 1: Energy value has been included in the manuscript

Response 2: Yes, the process is an established methodology for such review. reference has been added to the section. Already in the introduction as well, other studies in various fields aiming at sustainability that have employed such methods were also described and references provided.

Response 3: corrections have been effected

Response 4: Sentence in 146-147 has been paraphrased.

Response 5: Corrected

Response 6: Corrections have been effected

Response 7: Corrected

Response 8: Rephrased

Response 9: Spelling checked

Response 10: Corrected

Response 11: the sentence has been rearranged. section 3.4.1 describes figure 7, while section 3.4.2 describes figure 8.

Response 12: the correction above has addressed this as well.

Response 13: the graph has been corrected

Response 14: All clusters were number, 7 in all.

Response 15: sentence rephrased

Response 16: sentence rephrased

Response 17: spelling checked

Reviewer 2 Report

I know that green refuse-derived fuel is topical and I like the idea to use bibliometric analysis techniques. In the Introduction, I recommend stressing more the usefulness of your approach. Then I suggest rewriting the Methods section to be more precise and understandable, improving comments of results, and if possible rewriting section this section. From the literature and implications point of view, the authors could work a bit harder to provide a more specific picture of RDF and how it would help future academicians and practitioners. The interpretation of results could be explained with the perspective of future scope and also, the authors could add their personal insights in the result section. Lastly, enrich the literature add implications, and proofread the manuscript. I wish the very best to the authors for enriching the current piece of work.

Author Response

Response: Recommendations were noted and all have been affected in the main manuscript which is uploaded. The approach (bibliometric analysis) is an accepted method for literature review especially literature with a broad area to consider. References have been provided to support the steps used, while works that also utilized the approach have been provided in the introduction. The Introduction has been improved with more literature and implications. The results section has been updated with other relevant aspects that were not included initially and discussed with other studies as well

Reviewer 3 Report

Reviewer 1:

I recommend major amendments at this level.

General comments:

The manuscript entitled Bibliometric analysis; Characteristics and Trends of Refuse Derived Fuel Research.” was reviewed. The work carried out in the manuscript is interesting and aimed at a review analysis and an overview of the literature 109 on RDF from the Scopus database. However, the authors are suggested to undergo several major corrections as per the reviewer comments to improve the quality of the manuscript. Better connect your research findings to previous works published in Sustainability and in other top journals. The innovation and the importance of this work are not clearly highlighted in the abstract, introduction and conclusions. Please work on this and prove to us why this work is valuable. Would you explicitly specify the novelty of your work? What progress against the most recent state-of-the-art similar studies was made? Additionally, the novelty of the research still is not clear and the discussion and conclusions can not satisfy me. I recommend major amendments to the manuscript in the present form. Please provide research highlights to convey the core findings and provide readers with a quick textual overview of the article. It is highly recommended to provide a graphical abstract, as it will increase the visibility of the work and make the manuscript more appealing. The journal's author guidelines and instructions should be followed in preparing the revised version. Other main remarks that in my opinion needs attention are the following:

Detailed comments:

Title: Ok.

Abstract:

The abstract should state briefly the purpose of the research, the principal results and major conclusions. An abstract is often presented separately from the article, so it must be able to stand alone. The abstract should include a sentence about your findings, discussions and conclusions in your abstract and underscore the scientific value-added of your paper in your abstract. In the abstract, please add an indication of the achievements from your study that are relevant to the journal scope. Please be concise - maximum 1-2 lines.

Introduction:

The review of literature needs more updating with more recent references/works to have a clear and concise state of the art analysis. This should more clearly show the knowledge gaps identified and link them to the paper goals. Please remove all the lump of references. After that please check the manuscript thoroughly and eliminate all the lumps in the manuscript. This should be done by characterising each reference individually. This can be done by mentioning 1 or 2 phrases per reference to show how it is different from the others and why it deserves mentioning.

Materials and Methods:

The materials and methods have been written in sequence and structured well.

Results and Discussion:

All the findings of the current work need to be compared and discussed with the results of other researchers finding instead of having a general comparison with other researchers works. The authors should perform a comparison between the forecasting results. In your discussion section, please link your empirical results with a broader and deeper literature review.

Conclusions:

Too wordy, make it short and concise. The conclusion section appears to be just a detailed summary of results/observations. All conclusions must be convincing statements on what was found to be novel, impactful based on the strong support of the data/results/discussion. Please make sure your conclusions' section underscores the scientific value-added of your paper, and/or the applicability of your findings/results. Highlight the novelty of your study.

References:

Please check the reference section carefully and correct the inconsistency.

Author Response

Response 1 (Abstract): 

the abstract has been updated/modified accordingly in the manuscript. The purpose of the research is provided in the abstract introduction (lines 13-16). The findings from the bibliometric study are presented and discussed in the abstract (from line 21), and the conclusion as well. The achievement of the study is the response to the study objective which has given an overview of the research area of RDF and shared insights of the emerging and possible future focus. This is also a major premise for waste to energy as a sustainable waste management method, in line with the scope of the journal.

Response 2 (Introduction); : Comments addressed. The literature has been updated. The knowledge gap and goals have been stated clearly

Response 3 (Results and discussion): Suggestions are considered and corrected in the manuscript

Response 4 (Conclusions): the conclusion responded to the objectives set for the study, followed by a discussion of what the future research can address based on the major findings. The novelty of the review provided a general overview of RDF research and suggested specifics of the research area that has not been explored/ fully explored yet/improved (location, method, process).

Response 5 (Reference): The reference followed the sustainability style recommended. However, in addressing other reviewers’ comments, the reference list was updated and some corrections were effected. Hopefully, that corrects this as well.

Round 2

Reviewer 3 Report

Reviewer 2:

I have reviewed the revised version entitled" Bibliometric analysis; Characteristics and Trends of Refuse Derived Fuel Research". The paper has been improved and can be accepted. I do not have further comments.